# Developing a Card Game for Assessment and Intervention in the Person and the Family in Palliative Care: “*Pallium Game*”

**DOI:** 10.3390/ijerph20021449

**Published:** 2023-01-13

**Authors:** Carla Sílvia Fernandes, M. Belém Vale, Bruno Magalhães, João P. Castro, Marta D. Azevedo, Marisa Lourenço

**Affiliations:** 1Nursing School of Porto, Center for Health Technology and Services Research (CINTESIS), 4200-450 Porto, Portugal; 2Nursing in Hospital da Luz-Póvoa de Varzim, 4490-592 Póvoa de Varzim, Portugal; 3Department of Surgical Oncology, Portuguese Oncology Institute of Porto (IPO), 4200-072 Porto, Portugal; 4Oncology Nursing Research Unit IPO Porto Research Center (CI-IPOP), Portuguese (IPO Porto) Comprehensive Cancer Centre (Porto. CCC) & RISE@CI-IPOP (Health Research Network), 4200-072 Porto, Portugal; 5School of Health, University of Trás-os-Montes e Alto Douro (UTAD), 5000-801 Vila Real, Portugal; 6Wecare Saúde—Continuous Integrated and Palliative Care Unit, 4490-492 Póvoa de Varzim, Portugal

**Keywords:** palliative care, games, recreational, terminal care, family, family involvement, patient–caregiver communication

## Abstract

Communication between the multidisciplinary team, the person, and the family in palliative and end-of-life situations implies, in most situations, a high negative emotional burden. Therefore, innovative strategies are needed to reduce it. The goal of this study is to describe the various stages of development and validation of a collaborative card game for people in palliative care and their families. Phase one is an exploratory study, Phase two is a Delphi study, and Phase three is a multiple case study. Participants for phases 2 and 3 were recruited using a convenience sampling method. The results demonstrate in an organized and structured way the different phases required to build a collaborative card game. The use of the game was found to be useful and effective. Four categories emerged from the content analysis of the open-ended responses: usability, evaluation tool, communication and therapeutic relationship, and meaning when using the game. A collaborative game in palliative care helps to create a space for individuals and families to express feelings and experiences, meeting the myriad of physical, psychosocial, and spiritual needs. The “Pallium game” is a useful and impactful approach to discussing sensitive topics in palliative care.

## 1. Introduction

Palliative care can improve the quality of life of people and their families when timely and holistic identification of needs is carried out, allowing for the planning of person-centered care that meets their multidimensional needs and includes their wishes [1,2]. Palliative care should meet the physical, emotional, spiritual, and social needs of the sick person and those closest to them, accepting death as a part of life [3,4].

The terminal condition cannot be measured in terms of time and is instead considered a condition that comes from a serious, incurable, and uncontrollable disease, which can be experienced in hours, days, weeks, months, or years [5]. In this way, palliative care is applicable not only in the last stages of life but also from diagnosis to mourning [4,6]. Palliative care aims to offer a better quality of life, and care should be extended to family members and caregivers. It aims to prevent and relieve suffering through the early identification of needs, thus constituting an interdisciplinary field of total, active, and comprehensive care, offering psychosocial and spiritual support, including during the period of family grief [7].

When integrating a systemic view of the family, it is understood that the disease influences the whole family in one way or another [8]. It is difficult for just one family member to be aware of everything that happens within the family; therefore, it is crucial to adopt a family perspective so that health care facilitators can better meet the specific needs of each family [9]. Clear, in-depth communication provides insight into individual and family preferences for end-of-life care and treatment, enhancing understanding and shared decision-making [10,11].

It has been shown that tools in the form of cards with images or statements make it easier to put thoughts and feelings into words, particularly to discuss end-of-life issues [8,9]. In the context of palliative care, starting a conversation about certain themes is widely acknowledged as being difficult [12]. Collaborative games, particularly card games, have emerged as an ethical and viable strategy for a patient- and family-centered approach that is structured and organized to evaluate and intervene in the multiple needs of the person and the family in a palliative situation [2].

Despite the growing popularity of games in the health area, studies on games applied in the context of palliative care are still scarce [2]. However, card games in palliative care can promote quality of care [13]. The use of card games not only allows for participation in the game without any inhibitions and with a high degree of satisfaction but also allows for the discussion of sensitive topics related to the end of life, motivating participants to engage in behaviors related to advanced care planning [2].

Considering all the above, it seemed relevant to contribute to knowledge in this area. As a result, we set out to develop a card game for palliative care assessment and intervention in the individual and family.

## 2. Material and Methods

This article describes the three stages of development and validation of a collaborative card game for assessment and intervention in the person and family in palliative care: an exploratory study, a Delphi study, and a multiple case study.

### 2.1. Study Procedure and Participants

In the first phase, an exploratory study was carried out to identify the thematic areas for the cards over the next six months. Next, issues were explored further, such as studies on the use of games, family experiences in palliative care, individual and family needs in palliative care, instruments used in assessment, and individual and family intervention in palliative care.

In the second phase, a Delphi study was carried out to assess the questions to be included in the game cards. Based on the Delphi study methodology, we brought together a panel of experts in the fields of oncology and palliative care. Using a non-probabilistic sampling technique and a professional network (snowball), 100 experts were obtained who met the following criteria: a health professional working in oncology and palliative care with at least 2 years’ experience.

Next, based on evidence from previous literature and considering the various domains of palliative care, the game’s questions were constructed. Afterwards, they were sent by email, using a Google Forms^®^ questionnaire, to the panel of experts for a consensus study. The questionnaire consisted of two parts. The first part included the explanation of the study, free and informed consent, and the experts’ sociodemographic data. The second part consisted of 96 items divided into two areas: 72 questions and 23 interventions. Finally, the experts were asked to indicate their level of agreement regarding the relevance of the questions and interventions to be included in the card game. The experts were required to respond to the questionnaire within a defined time period. After analyzing and appraising the experts’ answers, as well as any suggestions, some questions and interventions were reformulated, others were added, and others were eliminated. Once the analysis was finished, we sent out the questionnaire for a second round. In the end, we reached consensus and definitively validated the items to integrate into the game. As a result of this step, the prototype of the card game was built, comprising 84 questions and 8 interventions. This prototype was later validated.

The data obtained from the questionnaires in the two rounds were submitted to descriptive statistical analysis, namely frequency, mean, and degree of agreement (in percent) with the question and the proposed intervention. Regarding the level of agreement, a percentage value of 75% was adopted as the required level of agreement, and 100% was considered a perfect consensus.

In the third phase, to validate the card game “*Pallium Game*”, we used the methodology of multiple cases to apply and validate the game with health professionals, patients, and families. The game was tested and validated in a palliative care unit, for which we obtained prior authorization from the institution and the competent authority (No. CE/2021/27). We created a form for the person, family, and health professional who accompanied the session to evaluate the instrument for assessment and intervention in the person and family in palliative care as a result of this stage and after using the game prototype.

The sample consisted of six cases (patients and families) with illnesses requiring admission to the palliative care unit and four health professionals (a nurse, a physician, an operational assistant, and a psychologist) who accompanied the implementation of the game. The inclusion criteria for integration in the validation phase were: people over 18 years of age, hospitalized in the palliative care unit (and/or their family members), who were alert and oriented with decision-making capacity, and who voluntarily agreed to participate in the study by signing informed consent. Participants were first approached by the unit’s psychology service and then referred for enrolment in the study if the person and family gave their consent.

We built a separate form for the person, the family, and the health professional to evaluate the game. This form was applied at the end of each session and was constructed with open and closed questions to collect sociodemographic data, game evaluation, advantages, disadvantages, difficulties, and needs. The closed questions consisted of Likert-type options from 1 to 5. For the analysis of the closed questions, a descriptive statistical analysis was performed, and, for the open questions, a content analysis was performed according to Bardin [14].

### 2.2. Ethical Considerations

In the Delphi study, informed consent was obtained from the group of experts before filling out the questionnaire. In the validation phase, voluntary informed written consent was obtained from the person and their family after a verbal explanation of all stages of the investigation. The document was previously made available for reflection. It was made clear that the person and family could refuse participation at any time during the investigation. The consent was signed in duplicate: one document for the person and family and one for the investigator. Likewise, for the participation of the health professional, consent was also obtained, guaranteeing the anonymity and confidentiality of the data. The consent was signed in duplicate and under the same conditions as the previous one. Throughout the duration of the study, the anonymity of the participants was maintained, and the confidentiality of the data obtained was guaranteed. In the data analysis, data encoding was used to guarantee its confidentiality. This codification is known exclusively to the researchers. In addition, security measures were implemented to protect the information.

## 3. Results

### 3.1. First Phase: Exploratory Study

The contents and conceptualization of the thematic areas to be included in the game resulted from previous studies based on the theoretical subsidies of several authors (Calgary’s Family Assessment and Intervention Model [15], Resilience Model [16], Theory of Transitions [17]), as well as various instruments for the assessment of symptoms in palliative care, scales of impact of symptoms on the quality of life of the person and the family, instruments on the burden of the caregiver and family in palliative care (ECCP-Palliative care capacity scale [18], EORTC QLQ—European Organization for Research and Treatment of Cancer Quality of Life Questionnaire [19], ESAS-Edmonton Symptom Assessment System [20], FACIT-F—Functional Assessment of Chronic Illness Therapy-Fatigue [21], GCQ—General Comfort Questionnaire [22], FS3I—Family systems stressors strengths inventory [23], MSAS-SF—Memorial Symptom Assessment Scale short form [24], POS-Palliative Care Outcome Scale [25]), and use of games in palliative care [26].

Based on these data, we designed an initial questionnaire with 72 questions and 23 interventions, integrating the card themes of the game proposal: family assessment, symptoms, impact, support, experiences, and interventions.

### 3.2. Second Phase—Delphi Study

In the first round, the questionnaire was sent by email to 100 experts, to whom 43 experts responded (43% response rate), and this took place from September to December 2020. Regarding the characterization of the experts in the first round, they were mostly female (86%), with an average age of 41.86 years. With regard to academic degrees, 42% were degree-level graduates, 49% had a master’s degree, and 9% had a doctorate degree. All experts had experience in the field, with an average of 17 years of experience. These professionals had experience in direct care (79%), professional training (9%), health service management (5%), and other contexts (7%).

One question and two proposed interventions were not agreed upon among the 72 questions and 23 interventions proposed in the first round. The average degree of agreement obtained was 90.63 (maximum value 100% and minimum value 67.4%). Some questions and interventions were reformulated in response to expert feedback, and 23 new items were added.

A new form was sent again by email for a second round with 118 items, which took place between February and April 2021. In this second stage, 29 experts of the 43 who had participated in the first phase responded. Regarding the characterization of these participants, they were mostly female (79%), with a mean age of 43 years. With regard to their academic degree, around 62% had a master’s degree, 24% had a bachelor’s degree, and 10% had a doctorate degree. All the participating experts had experience in the field, with an average of 23.55 years of experience.

In the second round, 17 questions and 8 interventions were proposed for elimination as their themes were similar. After removing them, the degree of agreement was on average 92.19 (with a maximum value of 100% and a minimum value of 75.86%). This stage also included the contributions of experts in the reformulation of the items, ending this stage with 93 cards.

### 3.3. “Pallium Game”

A card game, called the “Pallium game”, was developed to enhance the creation of a favorable environment for exposing feelings, experiences, and meaning and promote the discussion of goals, values, and preferences for decisions related to care at the end of life. This game is also intended to improve communication between people, families, caregivers, and health professionals by creating, through the game, a space that enables the understanding of behavior patterns in the family system, extracts meaning from adversity, and promotes better interaction and adaptation between members of the system. In addition, the “Pallium game” hoped to facilitate communicational clarity, stimulate caregiver and family skills to face adversity, strengthen the family’s strengths by encouraging the active process of restructuring and growth, promote a collaborative resolution, validate or normalize emotional responses, and help reduce isolation.

After analyzing the results of phases 1 and 2, the prototype of the card game with 93 cards was created: 2 “start” cards, 23 “family” cards, 16 “support” cards, 10 “impact” cards, 23 “meaning” cards, 11 “belief” cards, and 8 “intervention” cards. The use of the “*Pallium Game*” must be mediated by a qualified professional. This game was created to be applied to the person and the family, integrating all family members who want to participate and even being used without the presence of the patient.

The “*Pallium Game*” begins with the two “Start” cards, serving as an “Icebreaker”.

Then, participants choose the cards according to the topic they want to respond to. They read the cards aloud and responded to the themes. Take turns reading the cards aloud and answering the respective themes. Players can choose to skip the questions if they wish. The contents of the card must be read aloud, and after the card is answered, it is put aside, and so on successively until the game ends. The game continues until 30 questions are answered, integrating at least two cards from each evaluation category (Table 1).

The game can be applied at different times of the health-disease transition and more than once. When the game is over, at least two intervention cards must be collected and discussed (Figure 1).

### 3.4. Third Phase: Multiple Case Study

To validate the advantages and disadvantages of its use, the “*Pallium Game*” was applied in a palliative care unit between July and September 2021. The game was used with three palliative care patients, twelve family members, and four health professionals—nurses, physicians, and psychologists—after informed consent. It should be noted that this game was applied in different contexts: (a) only to the person with a palliative illness (due to the absence of reference relatives or those unable to be present); (b) to relatives without the sick person present because the clinical situation does not allow; and, finally, (c) to both the sick person and family members simultaneously. It should also be noted that it was applied in different locations—in the office and in the person’s room (an individual room)—due to the person’s physical incapacity to go to another place. The average duration of each session was about 45 min, although in families with several members, it was extended to about 60 min.

The socio-demographic characterization of “*Pallium Game*” users is shown in Table 2. The sample consisted of 19 users: the sick person in a palliative context (*n =* 3), family members (*n =* 12), and health professionals (*n =* 4). The family relationship or the provision of care to the sick person hospitalized in the institution is shared by all stakeholders. Regarding sociodemographic data, in our sample, all sick people were female (*n =* 3), with an average age of 69 years and an average hospital stay of 27 days.

Regarding family members, they were also mostly female (*n =* 8), with an average age of 49 years, and all with a direct degree of kinship with the sick person (spouse/children/daughter-in-law/son-in-law/granddaughter) (*n =* 12). The participating health professionals were also mostly female (*n =* 3), with an average age of 39 years and an average of 7 years of professional experience.

The six cases that were part of this validation phase are presented below to help better understand each family’s context and typology; we also present the family genogram of each case (Figure 2). In order to guarantee anonymity and confidentiality, the participants are coded as follows: sick person (D: D1 to D3), family member (F: F1 to F12), and health professional (P: P1 to P4).

In case 1, the following were included: 1 person with palliative disease (D1) was hospitalized in the unit for about 1 month, and a family member (his daughter) was also his caregiver (F1). The session was accompanied by Professional 1 (P1). In case 2, two family members participated: the husband (F2) and the daughter (F3) of the sick person hospitalized in the institution. The session was accompanied by Professional 2 (P2). In case 3, there were two family members: the wife, who was also the caregiver (F4), and the daughter (accompanied by her 8-month-old daughter) (F5) of the sick person admitted to the institution. The session was accompanied by Professional 1 (P1). In case 4, only the sick person in a palliative situation (D2), admitted to the institution for about 1 month, participated. The session was accompanied by Professional 4 (P4). In case 5, three family members participated: the wife, also the caregiver (F6), the son (F7), and the daughter-in-law (F8) of the sick person hospitalized in the institution. The session was accompanied by Professional 3 (P3). In case 6, the participants were: the sick person in a palliative situation (D3), who was hospitalized in the institution for about three weeks; the daughter (F9), the son (F10), the son-in-law (F11); the granddaughter (F12); and the 2-year-old great-granddaughter, who was also present. The session was accompanied by Professional 1 (P1).

From the evaluation, it was possible to verify that no sick person and/or family member in our sample had previous contact with this type of game. Only one health professional reported having had prior contact with this type of intervention. Participants rated the game as “very useful” (average = 4.8). There were no reported difficulties or obstacles related to the use of this instrument (*n =* 18), with the exception of one participant (*n =* 1) who reported having difficulty understanding the language of the questions: “I need some explanation to know what they mean” (D2). All participants reported that there was no need to address other topics in the questions in addition to those addressed. All participants agreed that using this instrument can allow health professionals to provide better care to the sick person and their family (average = 4.9).

Four categories emerged from the content analysis of the open answers: usability, assessment instrument, communication and therapeutic relationship, and meaning when using the game (Table 3).

As a result of the analysis, we can see that the participants refer to the game as “*easy to use*” (F2, F3, and F9), simple, and objective, referring to not feeling any hindrance or recognizing any disadvantage when applying it. Regarding the game as an assessment instrument, the answers point toward the game as a facilitator, allowing for a more accurate idea of the difficulties and needs of each family: “*It fully enhances the identification of needs*” (P1; P2).

All participants and, more specifically, health professionals refer to the facilitation of communication and the therapeutic relationship as an advantage of using this instrument: “*It allows you to start conversations and therapeutic interventions more spontaneously*” (P2), facilitating proximity with the person and family.

By analyzing the answers, we can verify the various meanings through the use of the game. The “*Pallium Game*” was referred to as an instrument that facilitates dialog and sharing—“It helps to verbalize some things that are inside that we never say” (D3, F10, F11)—and leads to reflection and self-awareness of situations—“Reflection on issues, which would not be done otherwise” (F4, F5).

We found that the emotional charge was very strong during the sessions when applying this game, and some issues were painful. The statement “*very strong emotional charge; it is not always easy to manage this*” (F12) demonstrates this. In this situation, space and time were provided for participants to manage their emotions and feelings, providing emotional support through listening, sharing, and empathy.

## 4. Discussion

### 4.1. Game Development

To develop this collaborative game, it was necessary to proceed with a structured design consisting of several stages with well-defined objectives to achieve the proposed results. In general, games must be carefully planned in order to achieve the expected results [8,11].

The choice for a card game results from the conviction that it is an appropriate instrument for the population under study, leading players to discuss their views on death and issues related to end-of-life planning, as mentioned by other authors [27,28,29].

It is described as an ethical and viable method to start conversations about desires and priorities at the end of life [1]. The name “Pallium”, derived from the Latin word for cape, cloak, or to cover up or cover, conceptualized the context of palliative care as alleviating symptoms with the primary goal of promoting the person’s comfort [30].

In palliative care, the person and the family need open and effective communication in order to meet the care goals and treatment preferences of the person, family, and caregiver [11,31]. Indeed, communication is the central theme of the “*Pallium Game*”. This card game is intended to serve as an initial step to clarify the goals of care and eliminate barriers so that we can discuss and document the values and preferences of the person’s care and also gather perspectives from the family.

With the understanding that the first conversations on certain topics are difficult and must be introduced with sensitivity, exploring the understanding of people’s feelings and emotions requires a significant time investment by the health professional and can often feel oppressive for the person and their family [10,29]. The use of a properly developed game can be an advantage for the suave professional to assess the needs of patients and caregivers and define strategies and means to meet them. The themes included in the game and verified by the experts go against what is expected as the objective of palliative care, namely the management of distressing symptoms, affirming life, and facing death as a normal process, integrating the psychological and spiritual aspects of care provision, offering a support system to help the person live as actively as possible until death, and also helping the family to deal with the illness and bereavement [32]. Finally, as all the participants commented, there was no need to add or discuss other issues, which leads us to conclude that the topics included were necessary and pertinent.

### 4.2. Game Evaluation

The participants rated the “*Pallium Game*” as very useful, and no difficulties or obstacles were reported related to its use, which is in line with other authors’ use of cards in this context [26,27,28]. Some authors show that the use of games with sensitive topics like these did not increase participants’ anxiety [11,27,28,29,30,31,32,33].

Regarding the views of the participating professionals and according to the results obtained, the participants agree that the use of this instrument can allow health professionals to provide better care to the sick person and their family, promoting better communication. This game can allow for better efficiency in communicating bad news and in planning care, in addition to avoiding situations that avoid confrontation with the family [2,26,27]. As we were able to demonstrate in a previous review study on the topic, the use of a card game in palliative care ensures greater involvement and an increase in self-efficacy, in addition to enabling change strategies and balanced decision-making [2].

Regarding usability, the “*Pallium Game*” was considered easy to use, practical, and simple with objective questions. Moreover, no disadvantages or hindrances were reported with its application. According to other authors, although with different games, participants report high satisfaction with the use of a card game [12,26,27], preferring a game to other strategies in palliative care [33].

As an evaluation instrument, the “*Pallium Game*” was evaluated as a means of sharing feelings and emotions, an “unlocker” of conversations and difficult subjects, and a facilitator of dialog. As noted in other studies, a game creates a safe and non-threatening environment that supports sensitive conversations [27] and allows the person and family to find their own answers [33].

The meaning attributed to using this game is visible in the participants’ discourse in particular. Reference was made to unlocking or releasing (“*It makes us feel lighter*”), leading to reflection, self-analysis, and self-awareness of their fears and expectations (“*It contributes to the expression of feelings, fears, and fears in an almost unconscious way, as it is a playful instrument*”). The cards facilitate the process of transforming thoughts and feelings into words [26].

Despite the references listed in the debriefing, it is important to mention other relevant aspects when evaluating the application of this game, namely those resulting from non-verbal communication, which was a common feature after the sessions and where expressions, feelings, and gestures stand out. Non-verbal communication stands out in the context of reconciliation between family members, with hugs between family members (who in some situations had previous conflicts), expressions of affection and apologies, moments of shared tears, and touch. These are data that cannot be mirrored in this document, but they allowed researchers to verify the effectiveness of this resource. The words and looks exchanged with the investigators as a form of thanks were universal, emphasizing how important this moment had been for them. These are non-measurable aspects that had a very positive impact on the application of this instrument. We add that, at no time, did any of the participants express a desire to leave the session, nor did they refuse to answer any question. On the contrary, participants expressed a desire to prolong the session and repeat the application of the game.

This study has some limitations, namely the sample size and the convenience sampling strategy, and therefore the results should be interpreted with care when considering the transferability of the findings. Despite these limitations, we believe that this study has implications for the improvement of care practice and for research into palliative care, as it provides an assessment and intervention tool for the person and the family. Our results also suggest that there is value in continuing to evaluate and develop game-based interventions to increase involvement in palliative care. Future research will put the gaming intervention through a randomized clinical trial and validate its effectiveness in larger samples.

## 5. Conclusions

In summary, the “*Pallium Game*” is a promising instrument for the person and family’s assessment and intervention in the context of palliative care. It is essential to find strategies that eliminate some barriers to facilitate discussion and be able to document the values and preferences of the person and family. In this context, a game can overcome various barriers by reformulating these discussions. The card game acts as an ice-breaker for difficult conversations and a facilitator of sharing emotions. It also supports more effective communication between the person, family, and professional, promoting an effective therapeutic relationship and facilitating conversations around difficult subjects.

We are aware that studies on this subject are still very scarce and have little scientific validity, and therefore it is necessary to invest in and further develop this area. During our journey, we encountered some challenges due to the limitations imposed by the current pandemic.

## Figures and Tables

**Figure 1 ijerph-20-01449-f001:**
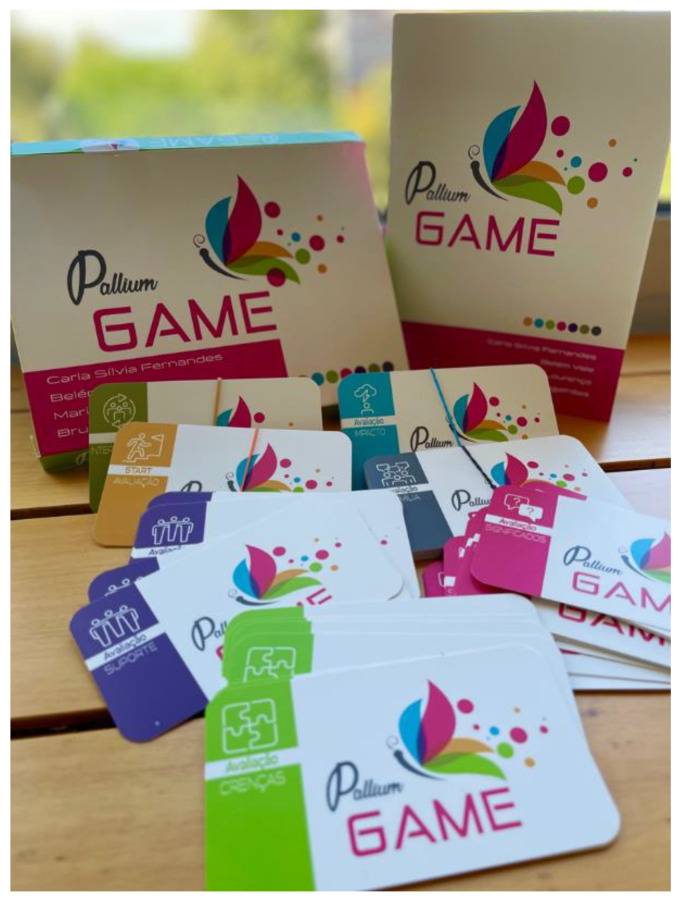
“*Pallium Game*”.

**Figure 2 ijerph-20-01449-f002:**
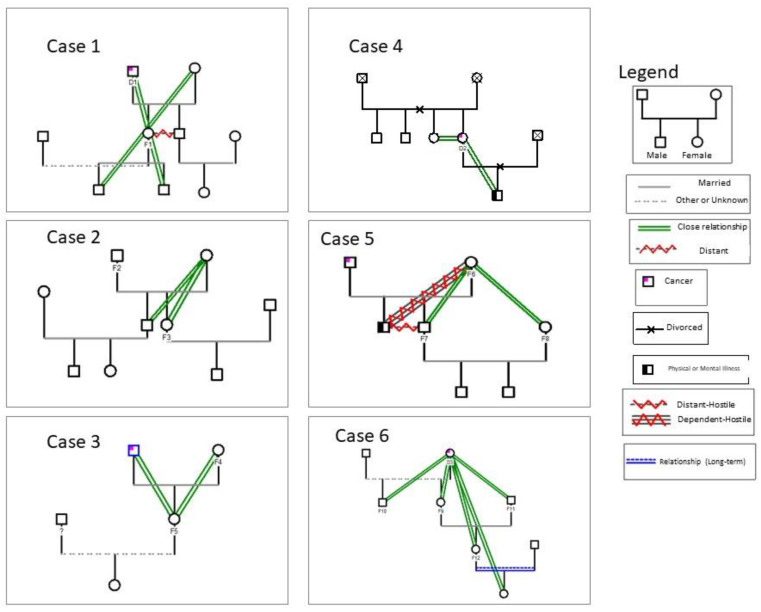
The family genogram of cases included in this study.

**Table 1 ijerph-20-01449-t001:** Examples of Category and Questions that make up the “*Pallium Game*”.

Category	Questions
**“Start” Cards**	− Of the people who make up the family, is there anyone who is not biologically related? (Significant people/pets?)
	− Can you tell me who is part of your family?
**_________________________**	____________________________________________________
	− At this moment, how do you see your body image?
**“Impact” Cards**	− How did the disease interfere with your monthly budget?
	− Have your symptoms been impeding your relationship with your close family and friends?
**_________________________**	____________________________________________________
	− Does your knowledge about the disease help or hinder your ability to cope?
**“Belief” Cards**	− How does your spiritual belief help you to understand the meaning of life and your existence at this stage?
	− What do you imagine the future to be like?
**_________________________**	____________________________________________________
	− Of the family members that live with you, who usually gives you the most support?
**“Family” Cards**	− Do you believe the disease has brought or separated you from your family members??
	− How have family members adjusted to the changes imposed by the disease?
**_________________________**	__________________________________________________
	− To what extent did other family members or close friends make themselves available to help you?
**“Support” Cards**	− How does information about the disease help you cope better with the situation?
	− How can health professionals contribute to your well-being and that of your family?
**_________________________**	____________________________________________________
	− At this stage of your illness, did you ever feel discouraged? Revolt? What do you do when you feel like this?
**“Meaning” Cards**	− Do you have the opportunity to discuss your fears and concerns with family and friends?
	− At this stage of your life, what is the most common feeling?
	− What is your main concern right presently?
**_________________________**	____________________________________________________
	− All gifts should praise each member of the family.
	− Make a list of three things that would make you feel more at ease right now.
**“Intervention” Cards**	− Plan an activity that you would enjoy doing as a family.
	− Do you have a problem with someone that you would like to solve?
**_________________________**	____________________________________________________

**Table 2 ijerph-20-01449-t002:** The socio-demographic characterization of “*Pallium Game*” users.

Variables	Results (Number of Responses) (*n =* 19)		
	Person in the Context of Palliative Disease (D) (*n =* 3)	Family (F) (*n =* 12)	Healthcare Professional (P) (*n =* 4)
**Genre**	Female *n =* 3 Male *n =* 0	Female *n =* 8 Male *n =* 4	Female *n =* 3 Male *n =* 1
**Age (mean)**	69 years	49 Years	39 Years
**Length of stay** **(mean)**	27 Days		
**Degree of kinship** (with the sick person)		Children *n* = 6 Spouse *n* = 3 Son-in-law *n* = 1 Daughter-in-law *n* = 1 Granddaughter *n* = 1	
**Professional activity**	Seamstress *n* = 1 Cartoner Operator *n* = 1 Merchant *n* = 1	Tailor *n* = 1 Engineering *n* = 2 Teaching/Science Education = 4 Tech. Administrative *n* = 2 Police = 1 Beautician *n* = 1 Civil Construction *n* = 1	Psychology *n* = 1 Nursing *n* = 1 Medicine *n* = 1 Operational Assistant *n* = 1
**Years of professional experience (mean)**			7 Years

**Table 3 ijerph-20-01449-t003:** Themes, categories, and registration units emerged from the content analysis of the open responses to the questionnaires.

Theme	Categories	Registration Unit
**Benefits** **and** **Disadvantages**	**Usability**	*“Practical” (D1, F4)* *“Simple and with objective questions” (D1, F4)* *“Easy to use” (F2, F3, F9)* *“Without any disadvantage” (D2, D3, F4, F5, F6)* *“I don’t see any disadvantages; it helps a lot.” (F2, F3, F7, F8, F9, F10, F11)*
	**Assessment Instrument**	*“Initiating conversation and therapeutic interventions more spontaneously” (P2)* *“Insert more or less susceptible subjects” (P2)* *“Exploring the most difficult, and sometimes addressing the impossible issues” (P2)* *“Facilitator of sharing emotions and feelings” (P3)* *“Sharing pain and feelings with the family in a less formal and easier way” (P4)* *“Easily identifies the needs of the user and family, allowing the development of an intervention more focused on the difficulties encountered” (P1)* *“Exploring more difficult and more difficult subjects to address” (P2)* *“Interaction with families in a lighter and more informal context” (P3)* *“More depth to define the current state” (P3)* *“Better knowledge of the general state of the family, how they react to pain, and what difficulties they encounter” (P4)* *“Yes—the themes presented on the cards can be used to verify the greatest difficulties of the family” (P1)* *“It fully enhances the identification of needs” (P1; P2)* *“It allows the identification of needs because it’s a real game” (P4)* *“It fully enhances the identification of needs due to the informality with which it is carried out” (P3)*
	**Communication and Therapeutic Relationship**	*“Facilitation of communication” (P1)* *“It supports professionals to establish the therapeutic relationship” (P1)* *“Closeness to family members” (P3)* *“More openness” (P3)* *“Enables more spontaneous conversations and therapeutic interventions.” (P2)*
	**The meaning of using the game**	*“Reflection on issues that would not be done otherwise” (F4, F5)* *“It helps to detach, unlock, and dialog” (D2, F12)* *“It makes the family talk” (F6)* *“It makes us talk about subjects we don’t have the courage to talk about” (F7)* *“It makes us feel lighter” (F8)* *“It helps to verbalize some things that are inside that we never say” (D3, F10, F11)* *“It was very useful” (F9)* *“It’s always good for us to deal with unknown emotions” (F12)* *“It can be very beneficial for the family, but a little painful for the patient” (D1, F1)* *“It has a very strong emotional charge; it is not always easy to manage this” (F12)* *“It’s a good way to express feelings” (F12)* *“Reflection of feelings” (P1)* *“It contributes to the expression of feelings, fears, fears in an almost unconscious way, as it is a playful instrument” (P1)*

## Data Availability

Not applicable.

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
