# Peer review of "Developing a Card Game for Assessment and Intervention in the Person and the Family in Palliative Care: “Pallium Game"

_ijerph, 2023, doi:10.3390/ijerph20021449_

Round 1

Reviewer 1 Report

Very interesting article, innovative and with a very high practical potential. I present some suggestions for improvement:

a) You could introduce the timeframe for the execution of the exploratory study;

b) in line 64 they refer that to start a conversation with certain topics in palliative care is difficult. It would be important to mention some studies that point out the difficult topics to talk about in palliative care.

c) The application of the Pallium Game has no a priori conditions, i.e. can it be applied to any patient in palliative care? And what about those patients who are transferred to palliative care and still do not know their clinical situation at all, can we apply the game?

d) We understood the advantages of the game in terms of communication, but I did not understand how the game helps in the planning of care as mentioned by the authors;

e) Finally, I think it would be important to show the face of the game, the cover or some of the elements that compose it.  

Author Response

Dear Elsie Wu

Thank you for reviewer A comments on the manuscript titled "Developing A Card Game for Assessment and Intervention in the Person and the Family in Palliative Care: Pallium Game". The comments are very valuable and useful for reviewing and improving our article, as well as of guiding importance for our research. We all do for the newspaper and hope that readers can understand our work in a fluid way. The reviewed gifts were marked in yellow.

Reviewer A
Present reviewed were marked in yellow.

(a) the time limit for carrying out the exploratory study could be introduced;

We added the time of the exploratory study in line 84.

... In the first phase, an exploratory study was conducted to identify the thematic areas of the cards, during six months.

b) In line 64 they state that starting a conversation with certain topics in palliative care is difficult. It would be important to mention some studies that point to the difficult topics to talk about in palliative care.

We mention an author who justifies the difficult themes to talk about palliative care, in line 64.

-Parry, R., Whittaker, B., Pino, M.et al. Real Talk evidence-based communication training resources: development of materials based on conversation analysis to support training in end-of-life health and social care conversations. BMC Med Educ 22, 637 (2022). https://doi.org/10.1186/s12909-022-03641-y

c) Does the application of the Pallium Game have a priori conditions, i.e. can it be applied to any patient in palliative care? And patients who are transferred to palliative care and still do not know their clinical situation, can we apply the game?

Yes, the game can be applied to all patients referred for palliative care and who are aware of their clinical situation. The themes covered in the game are focused on the field of losses to which people with advanced and irreversible chronic disease are subject. These losses are felt by the person and the family in all areas of the human, physical, psychological, social and spiritual condition. In the case of the conspiracy of silence towards the patient, this is a delicate situation. In our view, it should be an issue addressed by the multidisciplinary team responsible for the patient and family. The conspiracy of silence involves ethical issues that must be discussed by the team. Ultimately, this game can be used as a strategy to help communicate bad news during a family conference.

d) We understand the advantages of the game in terms of communication, but did not understand how the game assists in the planning of care as mentioned by the authors;

The Pallium Game is a strategy that facilitates the communication of issues that are difficult to address due to the emotional burden that may be involved in discussing end-of-life issues. From our experience of using the game with the patient and the family, we realized that, after each game, the data obtained clearly support the decision-making of the multidisciplinary team, in order to reduce the levels of psychological and spiritual suffering of these people.

See pages 336 to 345

e) Finally, I think it would be important to show the face of the game, the cover or some of the elements that compose it.

Face of Pallium Game, linha 227 a 229

Reviewer 2 Report

The abstract has to be revised, e.g., the first sentence is very hard to read. The second sentence starts with "this study" - whereas the reader does not know anything about the sudy yet. Typo in line 35 - you wrote Pallium (your own creation) with three L.

As for the paper in general:

From my point of view a major part is missing: The authors talk about how they developed the card game by also applying participatory research methods. BUT I really did not get any idea, what the card game is about. I have the impression the card game should/could support the communication process between patients and their care givers. Lines 210-223 provide some info, but not sufficient. No further details how to play the game are provided. I would advise also to include pictures of the card game and also the game instructions.

line 217: What do you mean with the term "letter" in that context. Pls clarify.

All the results (mainly Third Phase) (line 225-310) can not be assessed against the content / aim of the game.  

Figure 1: The legend needs more further explanation. As for some cases, the cases are not well readable. (Too many parts of the legend seem to overlap.)

Table 2 - heading: "registration unit": That term is not self explanatory, pls revise.

line 409: pls also include the country (you only mention "national funds"

The full sentence in line 128-130 could be deleted - is a "negative" repetition of the sentence above.

Author Response

Dear Elsie Wu

Thank you for reviewer B comments on the manuscript titled "Developing A Card Game for Assessment and Intervention in the Person and the Family in Palliative Care: “Pallium Game". The comments are very valuable and useful for reviewing and improving our article, as well as of guiding importance for our research. We all do for the newspaper and hope that readers can understand our work in a fluid way. The reviewed gifts were marked in yellow.

Reviewer B
Present reviewed were marked in yellow.

The abstract has to be revised, for example, the first sentence is very difficult to read. The second sentence starts with "this study" - while the reader still knows nothing about sudy. Typing error in line 35 - you wrote Pallium (your own creation) with three L's.

The abstract has been revised. Writing error has been corrected

Communication between the multidisciplinary team, the person and the family in palliative and end-of-life situations implies, in most situations, a high negative emotional burden. Therefore, innovative strategies are needed to reduce it. The aim of this study is to describe the different phases of construction and validation of a collaborative card game for the person and family in palliative situation. Phase 1- exploratory study, Phase 2 - Delphi study, and Phase 3 - multiple case study. Participants for phases 2 and 3 were recruited using a convenience sampling method. The results demonstrate in an organised and structured way the different phases required to build a collaborative card game. The use of the game was found to be useful and effective. Four categories emerged from the content analysis of the open-ended responses: usability, evaluation tool, communication and therapeutic-ship relationship, and meaning when using the game. A collaborative game in palliative care helps to create a space for individuals and families to express feelings and experiences, meeting the myriad of physical, psychosocial, and spiritual needs. The "Pallium game" is a useful and impactful approach to discussing sensitive topics in palliative care.

As for the article in general:

From my point of view, an important part is missing: the authors talk about how they developed the card game, also applying participatory research methods. BUT I really don't have any idea, what is the card game. I have the impression that the card game should/could support the communication process between patients and their caregivers. Lines 210-223 provide some information, but not enough. No further details are provided on how to play the game. I would also advise to include pictures of the card game as well as game instructions.

We've added a paragraph between line 72 and 75 that helps clarify the advantage of creating a card game. The use of card games not only allows for participation in the game without any inhibitions and with a high degree of satisfaction, but also allows for the discussion of sensitive topics related to the end of life, motivating participants to engage in advanced care planning behaviours [2].

Lines 219 to 227 describe some of the game rules.

Face of “Pallium Game”, line 229 to 232

line 217: What do you mean by the term "letter" in this context. pls clarify

The term letter was changed, by card, son line 219 and 220.

The “Pallium Game” begins with the two “Start” cards, serving as an “Icebreaker”. Then, participants choose the cards according to the topic they want to respond to. They read the cards aloud and respond to the themes. Take turns reading the cards aloud and answering the respective themes. Players can choose to skip the questions if they wish. The contents of the card must be read aloud, and after the card is answered, it is put aside, so on successively until the game ends. The game continues until 30 questions are answered, integrating at least two cards from each evaluation category. The game can be applied at different times of the health-disease transition and can be applied more than once. When the game is over, at least two intervention cards must be collected and discussed

All results (mainly Third Phase) (line 225-310) cannot be evaluated in relation to the content / objective of the game.

We changed the sentence in line 234. What we evaluated were the advantages and disadvantages of using the game, as exposed by the users.

To validate the advantages and disadvantages of its use, the “Pallium Game” was applied in a palliative care unit between July and September 2021.

Figure 1: Legend needs further explanation. As for some cases, the cases are not well readable. (Many parts of the legend seem to overlap.)

We changed the term Participants to “Pallium Game” users, on line 246.

The legend of Table 2 becomes: The socio-demographic characterization of “Pallium Game” users.

We changed the caption current of Figure 2:

Family Genogram of cases included in this study

Table 2 - Title: "Registration unit": This term is not self-explanatory, pls review.

Table 2 shows the result of the content analysis that emerged from the open responses included in the assessment instruments. Used to determine the advantages and disadvantages experienced by users of the “Pallium Game”.

The legend of Table 2 has been changed_ Themes, Categories and Registration Units, that emerged from the content analysis, from the open responses of the questionnaires.

line 409: pls also include the country (you only mention "national funds "

 Portuguese Funds

The complete sentence on line 128-130 could be deleted - it is a "negative" repetition of the sentence above

The sentence: “with illnesses requiring admission to the palliative care unit and four health professionals” and “The sentence: “People under 18 years of age, who were confused, did not have the capacity to make decisions, and people, families, and health professionals who did not agree to participate in the study were excluded”. They were removed because they were duplicates.

Round 2

Reviewer 2 Report

I would still be interested in more details of the card game, e.g. an overview of the 30 questions you mention.

Author Response

Thanks for your suggestion.
Table 1 shows the question categories and some sample questions that are part of the game. This was added on page 6 line 228. |
I hope in this way to have improved the clarity of presentation of the results of the manuscript.
